# Value of low-keV virtual monoenergetic plus dual-energy computed tomographic imaging for detection of acute pulmonary embolism

Yutthaphan Wannasopha[1], Kantheera Leesmidt[1], Tanop Srisuwan[1], Juntima Euathrongchit[1], Apichat Tantraworasin[2,3]*

1 Faculty of Medicine, Department of Radiology, Chiang Mai University, Chiang Mai, Thailand, 2 Faculty of Medicine, Clinical Epidemiology and Clinical Statistics Center and Department of Surgery, Chiang Mai University, Chiang Mai, Thailand, 3 Clinical Surgical Research Center, Chiang Mai University, Chiang Mai, Thailand

* apichat.t@cmu.ac.th, ohm_med@hotmail.com

**Data Availability Statement:** All relevant data are within the paper and its Supporting Information files.

## Abstract

### Objective

To compare diagnostic values between the 40 keV virtual monoenergetic plus (40 keV VMI+) dual source dual energy computed tomography (DSDECT) pulmonary angiography images and the standard mixed (90 and 150 kV) images for the detection of acute pulmonary embolism (PE).

### Methods

Chest DSDECTs of 64 patients who were suspected of having acute PE were retrospectively reviewed by two independent reviewers. The assessments of acute PE of all patients on a per-location basis were compared between the 40 keV VMI+ and the standard mixed datasets (reference standard) with a two-week interval.

### Results

This study consisted of 64 patients (33 women and 31 men; mean age, 60.2 years; range 18–90 years), with a total of 512 locations. The interobserver agreement (Kappa) for detection of acute PE using the 40 keV VMI+ images and the standard mixed CT images were 0.7478 and 0.8750 respectively. The area under receiver operating characteristics (AuROC) for diagnosis of acute PE using the 40 keV VMI+ was 0.882. Four locations (0.78%) revealed a false negative result. Hypodense filling defects were identified in twelve locations (1.95%) in the 40 keV VMI+ images but had been interpreted as a negative study in the standard mixed CT images. The repeated reviews revealed that each location contained a hypodense filling defect but was overlooked on the standard mixed CT images.

### Conclusions

Low-energy VMI + DSDECT images have beneficial in improving the diagnostic value of acute PE in doubtful or disregarded standard mixed images.

**Funding:** This study was supported in part by Chiang Mai University, Chiang Mai, Thailand in the form of a grant (Grant No. 17/2565) awarded to AT. No additional external funding was received for this study.

**Competing interests:** The authors have declared that no competing interests exist.

**Abbreviations:** the 40 keV VMI+, The 40 Kilo Electron Volt virtual monoenergetic plus; DSDECT, Dual source dual energy computed tomography; PE, Acute pulmonary embolism; CTPA, Computed tomography pulmonary angiography; VMIs, Virtual monoenergetic images; CNR, Contrast-to-noise ratios; MPA, Main pulmonary artery; RPA, Right pulmonary artery; LPA, Left pulmonary artery; RUL, Right upper lobe; RML, Right middle lob; RLL, Right lower lobe; LUL, Left upper lobe; LLL, Left lower lobe; ROI, Region of interest; FOV, Field of view; kVp, Kilovoltage peak; mAs, Milliamperage.

## Introduction

Acute pulmonary embolism (PE) is the third most frequent cause of death from acute cardio-vascular event, after myocardial infarction and stroke [1]. PE is a serious public health problem which results in thousands of deaths each year because it frequently goes undetected or has delayed diagnosis. The accurate diagnosis is vital to prevent fatal consequences.

Computed tomographic pulmonary angiography (CTPA) is the current standard modality of choice for diagnosing acute PE [2]. The image quality of CTPA has improved within the past two decades with consequent reductions in the percentage of non-diagnostic results [2, 3]. However, the suboptimal quality of images still occurs and leads to misdiagnosis or uncertain diagnosis. The indeterminate examinations from CTPA range from 0.03 to 10% [4, 5]. The major reasons for suboptimal vascular enhancement of CTPA have been referred to as incorrect bolus timing, delayed transit times, transient interruption of contrast material and low right ventricular cardiac output [6–9].

Improved contrast enhancement has been demonstrated by low-kilovoltage studies, as well as being a technique for reducing radiation dose [10–13]. Dual source dual-energy CT (DSDECT) became clinically available in 2006 [14]. DSDECT has two separate detector arrays which result in two different image datasets from two corresponding separate x-ray tubes (one with high kilovoltage, and the other one with low kilovoltage), located on the perpendicular rotating gantry. These two tubes can also be operated independently at different kilovoltage peak (kVp) and milliamperage (mAs) settings. Each dual-energy acquisition from a dual-source CT system automatically generates three separate sets of images: the high-energy (140–150 kVp) images, the low-energy (80–100 kVp) images, and the standard mixed dataset images [15, 16]. The standard mixed dataset images have an overall appearance equivalent to the traditional 120-kVp single-energy CT images. In addition, the image data series from DSDECT can be used to reconstruct additional virtual monoenergetic plus images (VMIs) at a chosen hypothetical energy level. The low-kilovoltage VMI has been reported to substantially increase the contrast enhancement of the pulmonary arteries, even to the subsegmental vessel level [17–20]. However, images reconstructed at very low keV levels were degraded by a drastic increase in image noise artifacts and thus supposed suboptimal interpretation. A noise optimized VMI algorithm or monoenergetic plus image (VMI+ or monoenergetic+) has been developed to allow increased iodine attenuation at low keV levels without the consequences of amplified image noise artifacts [21] (Fig 1). A merging of the high signal at lower energy datasets and the noise reduction at higher energy datasets have been performed to establish the monoenergetic plus images which revealed high contrast and low noise artifacts. Previous studies on the monoenergetic plus algorithm postulated the highest iodine contrast-to-noise ratios (CNR) at 40keV with a gradual reduction of increasing keV levels [21, 22].

In addition, the use of low-keV virtual monoenergetic plus images to provide optimal pulmonary vascular enhancement can allow for reduction of the contrast material volume while maintaining image quality [23]. These are advantageous for the patients with compromised renal function who are at risk of contrast-induced nephropathy and are also beneficial when there is a need for multiple iodinated contrast medium doses in a short time interval (<24 hours) [24].

However, only a few studies have reported on the clinical application for acute pulmonary embolism detection by using low-keV virtual monoenergetic plus images of DSDECT [25–27]. The objective of this study was to compare the diagnostic values between the 40 keV virtual monoenergetic plus DSDECT pulmonary angiography images and the standard dual-energy mixed (90 and 150 kV) images for the detection of acute pulmonary embolism in patients clinically suspected of having acute PE.

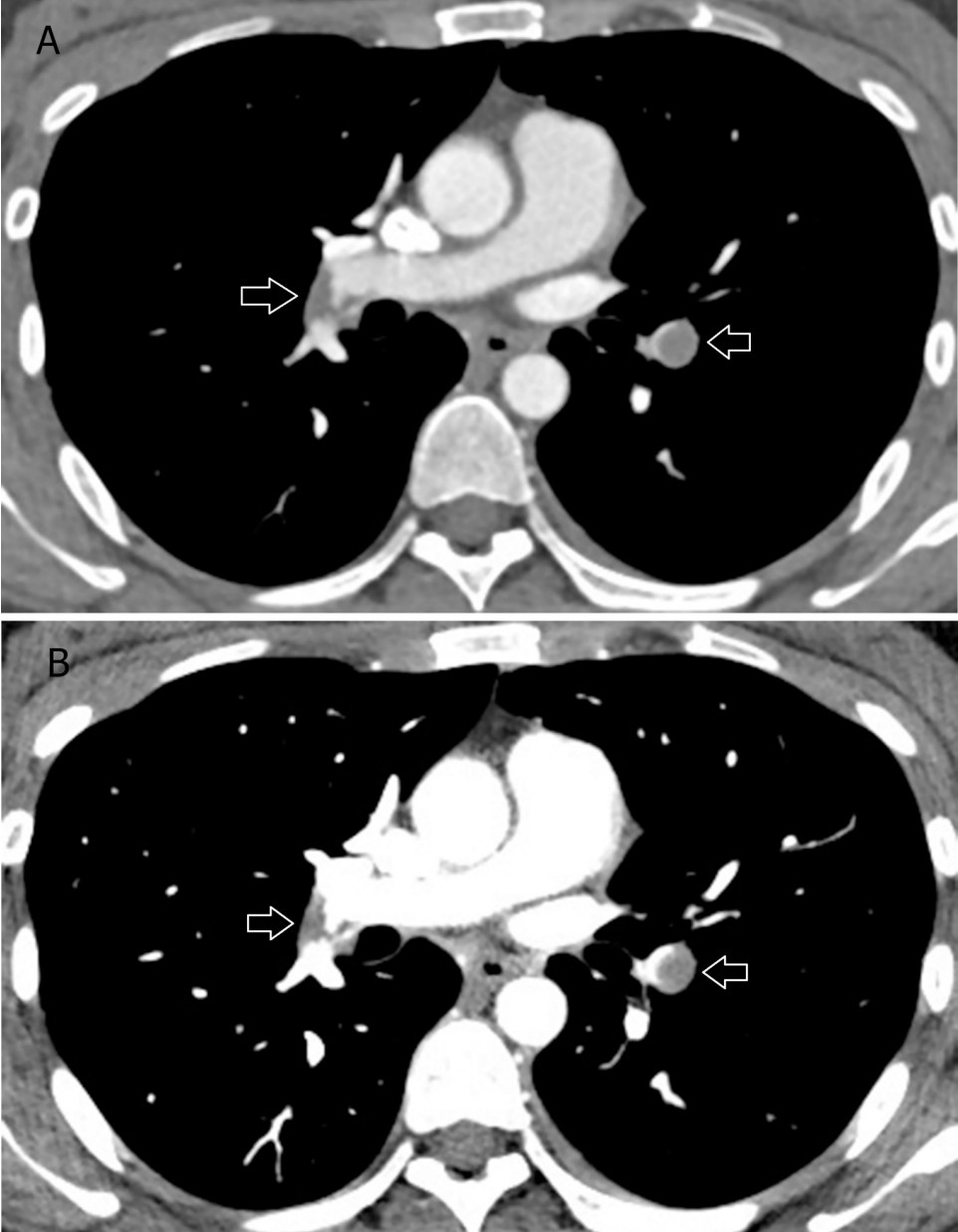

**Fig 1. Comparison between the standard mixed image and the low-energy VMI+ image.** A 65-year-old man with acute pulmonary embolism. DSDECT images at the level of the right pulmonary artery are shown in the standard mixed image (A) and in the low-energy VMI+ (B). Increased contrast enhancement of the vessels at the same CT window level and width is demonstrated in the low-energy VMI+. Hypodense filling defects of acute pulmonary emboli are also detected in the descending pulmonary arteries in the bilateral lower lobes. DSDECT, dual source dual energy computed tomography; VMI+, virtual monoenergetic plus.

## Materials and methods

### Patients and radiological evaluation

Institutional review board approval (Research Ethics Committee Faculty of Medicine, Chiang Mai University No: RAD-2560-05199) was given with a waiver for written informed consent due to the retrospective nature of the study. This research used data from patients who were

suspected of having acute PE and underwent DSDECT in our institution between September 2015 and October 2017. Exclusion criteria were as follows: age younger than 18 years, excessive motion artifacts, and suboptimal vascular enhancement due to technical error or cardiac status of the patients. Eventually, 262 patients were enrolled onto the study. From the 262 patients, there were 32 patients who had a CT report confirming acute PE. Then we randomly selected 32 patients who had a negative result for PE for the control study group. Therefore, the study population totaled 64 nonconsecutive patients (Fig 2). The radiologic images were extracted from the Chiang Mai University Picture Archiving and Communication System (CMU-PACS).

The distribution of acute PE was recorded by division of the pulmonary arteries into the 3 central locations; main pulmonary artery (MPA), right pulmonary artery (RPA), and left pulmonary artery (LPA); and 5 peripheral locations; right upper lobe (RUL) pulmonary artery,

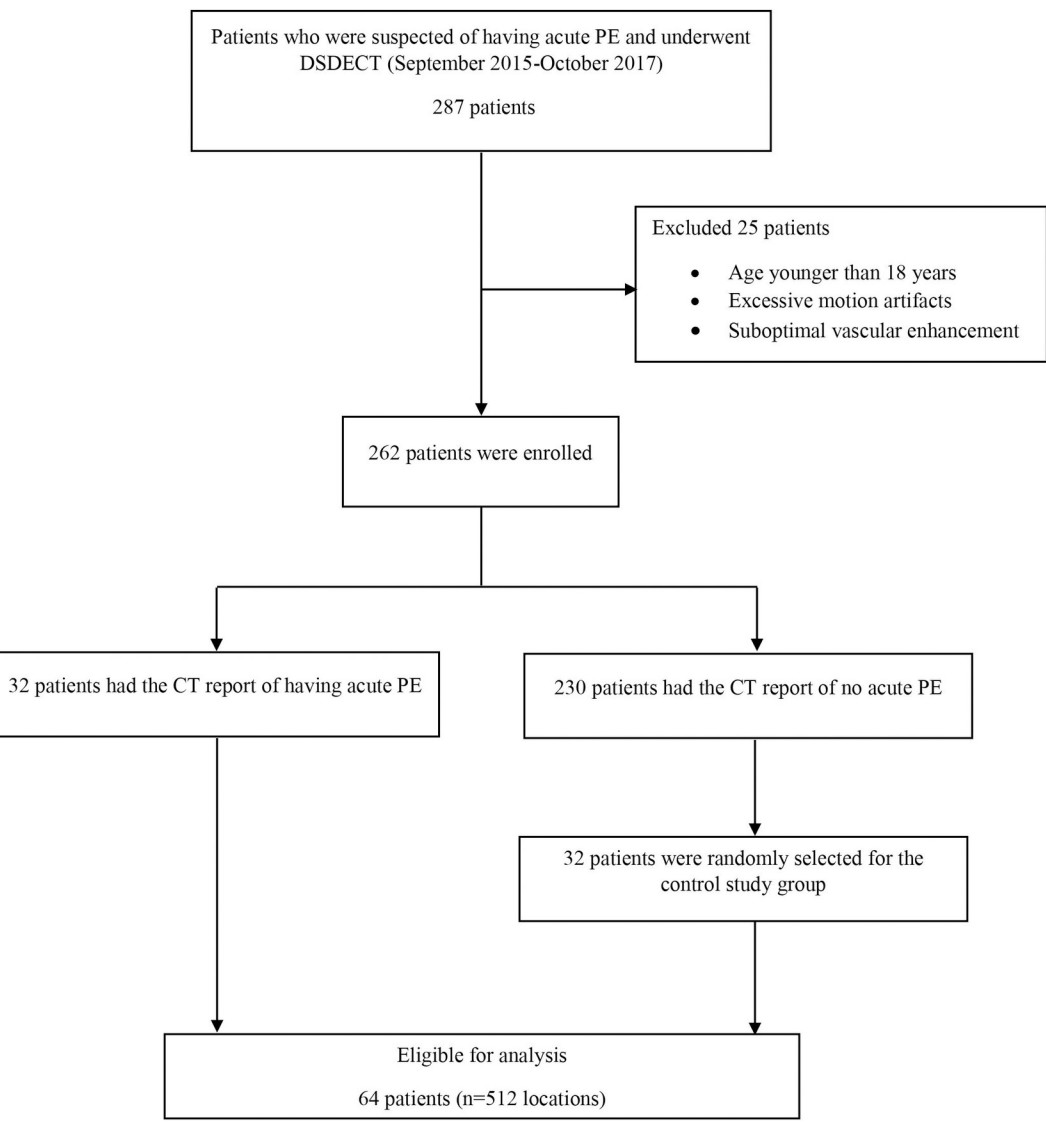

**Fig 2. Study workflow for patient recruitment.** PE, pulmonary embolism; DSDECT, dual source dual energy computed tomography.

right middle lobe (RML) pulmonary artery, right lower lobe (RLL) pulmonary artery, left upper lobe (LUL) pulmonary artery, and left lower lobe (LLL) pulmonary artery. Therefore, 512 locations (n = 512) from pulmonary arteries were evaluated by each of the two radiologists. The detection of acute PE in all locations was compared between the 40 keV VMI+ dataset and the standard mixed dataset which served as the reference standard for the detection of acute PE. The results were classified into positive, when the presence of an intraluminal hypodense filling defect was identified, and negative when there was no evidence of an intraluminal hypodense filling defect.

Chest DSDECT images of the study population were reviewed independently by two radiologists; Y.W. and T.S. who had nine years and fourteen years of experience in interpreting cardiothoracic CT respectively, on an offline workstation. In a blind evaluation, interpretations of pulmonary arteries for assessment of acute PE between the low-keV VMI+ (40 keV) and the standard mixed datasets were designed to be obtained separately with a two-week interval.

## CT protocol and image reconstruction postprocessing

The parameters of the dual-source scanner (SOMATOM Force; Siemens Healthcare, Forchheim, Germany) were as follows: tube, 2 x Vectron X-ray tubes (tube A, 90 kV/95-mAs and tube B, 150 kV/60-mAs); detector, 2 x Stellar detectors with 3D anti-scatter collimator; number of acquired slices, 384 (2 × 192) slices; rotation time, up to 0.25 s; in-plane temporal resolution, up to 66 ms; general power, 240 kW (2 x 120 kW); kV steps, 70–150 kV, in steps of 10 kV; and spatial resolution, 0.24 mm. Scans were performed in a craniocaudal direction, from the lung apices through to the lung bases, in suspended respiration. Scan initiating time was determined with the region of interest (ROI) placed on the main pulmonary artery. Fifty to one hundred ml (350 mg iodine/ml) of the contrast agent depending on the body weight of each patient at a dose of 1 ml/kg was injected at a rate of 4.0–4.5 ml/sec. The volume CT dose indices were recorded from the examination protocols. The images were reconstructed with 1-mm slice thickness and the field of view (FOV) that covered the entire thorax.

The standard mixed DSDECT datasets were automatically generated as linearly-combined images with a 0.6 blending factor which comprised of 60% 90-KV and 40% 150-KV and the overall appearances were equivalent to the traditional 120-kVp single-energy CT images. The additional virtual monoenergetic plus images (VMI+) were reconstructed at a chosen hypothetical energy level (40-keV) in axial view using a section thickness of 3 mm and an increment of 2 mm.

## Statistical analysis

The categorical data were reported as frequency and percentage. The interobserver agreement was reported as a kappa statistic. The kappa values were interpreted according to the Viera classification: 0.01–0.20 slight agreement, 0.21–0.40 fair agreement, 0.41–0.60 moderate agreement, 0.61–0.80 substantial agreement, and 0.81–0.99 almost perfect agreement [28]. Diagnostic index including sensitivity, specificity, area under receiver operating characteristic (AuROC), positive of likelihood ratio (LR+), positive predictive value (PPV), and negative predictive value (NPV) of the 40 keV VMI+ to diagnose acute PE were reported. The agreement between the 40 keV VMI+ and the standard mixed protocol to identify acute PE was evaluated by kappa statistic. A p-value of less than 0.05 indicated a statistically significant difference. All statistical analyses were performed using STATA program version 16.0.

**Table 1. Patient characteristics.**

| Variable | Positive | Negative | P-value |
|---|---|---|---|
| | n (%) | n (%) | |
| | N = 32 | N = 32 | |
| Weight (Median; IQR) (kg) | 60.3 (52.1–69.5) | 62.0 (53.6–69.8) | 0.941 |
| Height (Median; IQR) (cm) | 159.5 (155.0–164.8) | 160.0 (155.0–165.5) | 0.882 |
| BMI (Median; IQR) (kg/m$^2$) | 23.7 (22.0–25.7) | 23.2 (22.0–26.5) | 0.799 |
| BMI (subgroup) (kg/m$^2$) | | | 0.580 |
| Non-obesity (< 25) | 21 (65.6) | 23 (71.9) | |
| Pre-obesity (25–29.9) | 8 (25.0) | 8 (25.0) | |
| Obesity (≥30) | 3 (9.4) | 1 (3.1) | |

BMI, body mass index; IQR, interquartile range; kg, kilogram; cm, centimeter.

## Results

The study group consisted of 64 patients (33 women and 31 men; mean age, 60.2 years; range, 18–90 years) with clinically suspected acute PE who underwent DSDECT examinations of their pulmonary artery.

Patient characteristics and P values for comparisons between the two study groups based on the presence or absence of acute PE are listed in Table 1. The median weight and height of the positive PE group were slightly lower (60.3 kg vs. 62.0 kg; P = 0.941; and 159.5 cm vs. 160.0 cm; P = 0.882, respectively). In regard to BMI subgroup analysis, none of the differences were statistically significant.

Chest DSDECT of each location of the pulmonary arteries were reviewed independently by two radiologists. The level of agreement between the two radiologists revealed an almost perfect correlation (Kappa = 0.8750; P < 0.001) with a substantial correlation (Kappa = 0.7478; P < 0.001) in the standard mixed and the 40 keV VMI+ datasets, respectively (Table 2).

The detection of acute PE in the 40 keV VMI+ dataset was compared to the standard mixed CT images from all patients. The results were classified into positive and negative which revealed an almost perfect correlation between both datasets in the central pulmonary arteries (Kappa > 0.90) and a substantial to almost perfect correlation in the peripheral arteries (Kappa = 0.76–0.90) (Table 3).

Only one out of the 64 patients had a pulmonary embolism in the main pulmonary artery and the rest revealed negative results at this location. The results of the 40 keV VMI+ and the standard mixed images were concordant at the main pulmonary artery. At the right pulmonary artery, one individual was considered positive in the 40 keV VMI+ but negative in the standard protocol. At the left pulmonary artery, there was only one discordant study which was interpreted to be negative in the 40 keV VMI+ but positive in the standard protocol.

**Table 2. The interobserver agreement in interpretation of DSDECT image.**

| DSDECT image | Percent of Agreement | Kappa | P value |
|---|---|---|---|
| The standard mixed dataset | 93.75% | 0.8750 | < 0.001 |
| The 40 keV VMI+ dataset | 87.50% | 0.7478 | < 0.001 |

DSDECT, dual source dual energy computed tomography; the 40 keV VMI+, the 40 Kilo Electron Volt virtual monoenergetic plus.

**Table 3. Interpretation of dual source dual energy computed tomography according to the location: Comparison between the 40 keV VMI+ and the standard mixed CT images.**

| Location | The standard mixed image | | The 40 keV VMI+ | | Percent of agreement | Kappa |
|---|---|---|---|---|---|---|
| | Positive (percent) | Negative (percent) | Positive (percent) | Negative (percent) | | |
| MPA | 1 (1.6) | 63 (98.4) | 1 (1.6) | 63 (98.4) | 100.00% | 1.0000 |
| RPA | 14 (21.9) | 50 (78.1) | 15 (23.4) | 49 (76.6) | 98.44% | 0.9554 |
| LPA | 11 (17.2) | 53 (82.8) | 10 (15.6) | 54 (84.4) | 98.44% | 0.9431 |
| RUL | 21 (32.8) | 43 (67.2) | 19 (29.7) | 45 (70.3) | 90.63% | 0.7821 |
| RML | 21 (32.8) | 43 (67.2) | 20 (31.3) | 44 (68.8) | 95.31% | 0.8924 |
| RLL | 21 (32.8) | 43 (67.2) | 24 (37.5) | 40 (62.5) | 80.06% | 0.7607 |
| LUL | 18 (28.1) | 46 (71.9) | 23 (35.9) | 41 (64.1) | 92.19% | 0.8218 |
| LLL | 21 (32.8) | 43 (67.2) | 24 (37.5) | 40 (62.5) | 95.31% | 0.8974 |

The 40 keV VMI+, the 40 Kilo Electron Volt virtual monoenergetic plus; MPA, main pulmonary artery; RPA, right pulmonary artery; LPA, left pulmonary artery; RUL, right upper lobe; RML, right middle lobe; RLL, right lower lobe; LUL, left upper lobe; LLL, left lower lobe.

Peripheral pulmonary arteries were classified by lobes of the lung. In the right upper lobe, two locations revealed a positive finding in the standard mixed images but negative in the 40 keV VMI+ and the rest showed the same results in both protocols. In the right middle lobe, only one study was interpreted as positive in the standard mixed images but negative in the 40 keV VMI+ and the rest were concordant. In the right lower lobe, three locations were identified as positive in the 40 keV VMI+ but were reported as negative in the standard mixed images (Fig 3).

The left upper lobe pulmonary arteries revealed the most frequent mismatch of results between the two datasets. There were five locations that were interpreted as positive in the 40 keV VMI+ but interpreted as negative in the standard mixed dataset (Fig 4). In the left lower lobe, there were three locations were classed as positive in the 40 keV VMI+ but negative in the standard mixed images (Fig 5); however, the rest showed concordant results.

The diagnostic indexes including sensitivity, specificity, AuROC, LR+, PPV, and NPV of the 40 keV VMI+ to identify acute PE according to the location are displayed in the Table 4. According to the AuROC analysis, the 40 keV VMI+ showed very good performance for the diagnosis of acute PE in every location (AuROC = 0.882).

## Discussion

Many studies have proposed that the use of low-energy VMI+ can substantially increase the enhancement of the imaging of pulmonary arteries down to the subsegmental levels without a significant reduction in image quality which may increase diagnostic confidence for the detection of acute pulmonary embolism [17–19, 21–23, 27, 29, 30]. Our study was to verify this point of view. Our object was to compare the diagnostic values for the detection of acute pulmonary embolism in the patients who were clinically suspected of having acute PE, between the 40 keV virtual monoenergetic plus DSDECT pulmonary angiography images and the standard mixed (90 and 150 kV) dual-energy CT images which provided the standard of reference.

The study by Meier et al claimed that the use of VMI+ reconstructions from dual energy CTPA data at 40 keV is recommended for obtaining the best image quality of the pulmonary artery circulation [27]. Likewise, the 40 keV VMI+ technique can reduce the artifacts from transient interruption of contrast which is the contrast material dilution resulting from an intermixture between contrast material from the superior vena cava and a fluctuating amount of blood from the inferior vena cava.

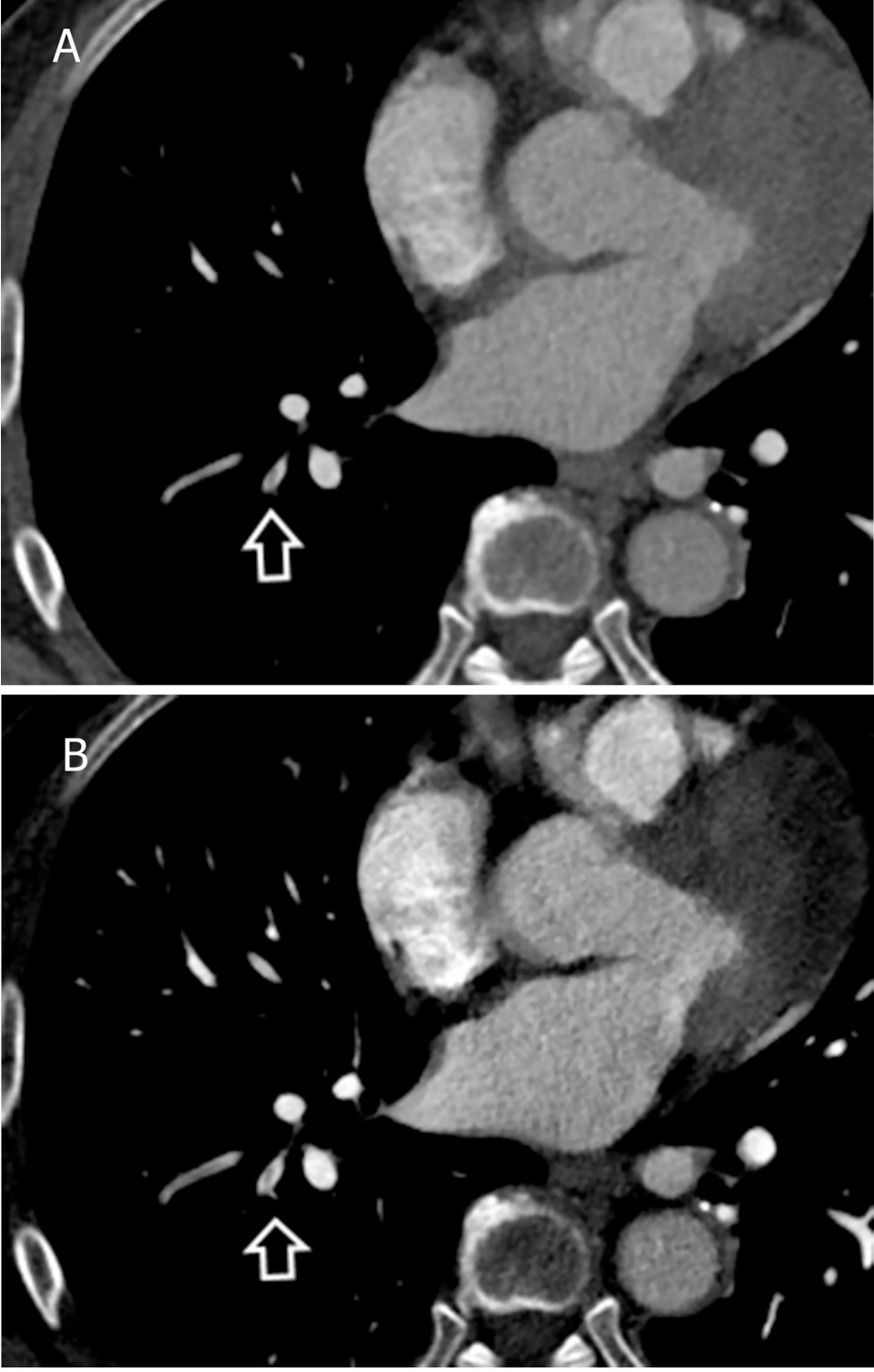

**Fig 3. Sample of an overlooked case.** A 22-year-old man with acute pulmonary embolism. DSDECT images at the level of the left atrium are displayed. A small right lower lobe segmental embolus (arrow) was unnoticed in the standard mixed image (A) but was identified in the low-energy VMI+ (B). DSDECT, dual source dual energy computed tomography; VMI+, virtual monoenergetic plus.

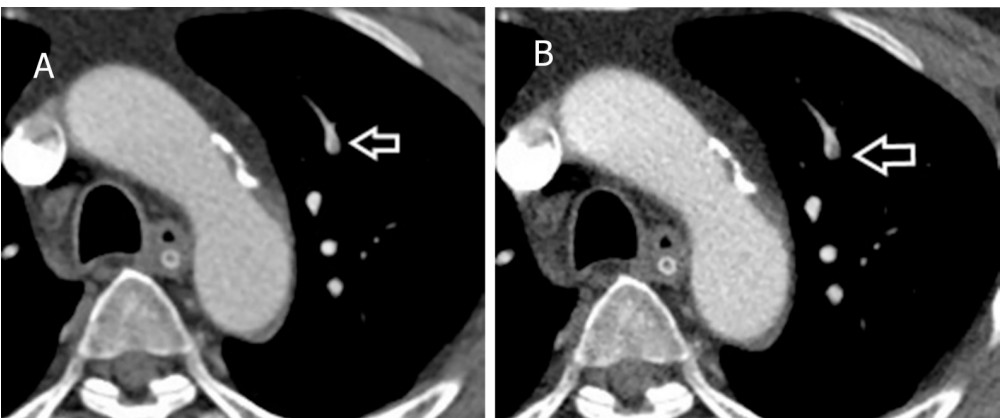

**Fig 4. Sample of an overlooked case.** A 76-year-old woman with acute pulmonary embolism. DSDECT images at the level of the aortic arch are displayed. A subtle left upper lobe segmental embolus (arrow) was unidentified in the standard mixed image (A) but was identified in the low-energy VMI+ (B). DSDECT, dual source dual energy computed tomography; VMI+, virtual monoenergetic plus.

In this study, there was substantial interobserver agreement (Kappa = 0.7478) in the detection of hypodense filling defects of pulmonary embolism by using the low-energy VMI+ while almost perfect interobserver agreement was found in the standard mixed dataset images (Kappa = 0.8750). These results imply that the minimal inferiority of subjective image quality with the low-energy VMI+ may affect the confidence of the reader in making an accurate diagnosis of pulmonary embolism. However, this would benefit from further validation in a larger study population. Because of the substantial to almost perfect interobserver agreement, we selected the data from one reviewer for further evaluation of the diagnostic index of the 40 keV VMI+ in identifying acute PE.

Our study revealed an almost perfect correlation between the detection of pulmonary embolism with the low-energy VMI+ and the standard mixed dataset CT images in the central

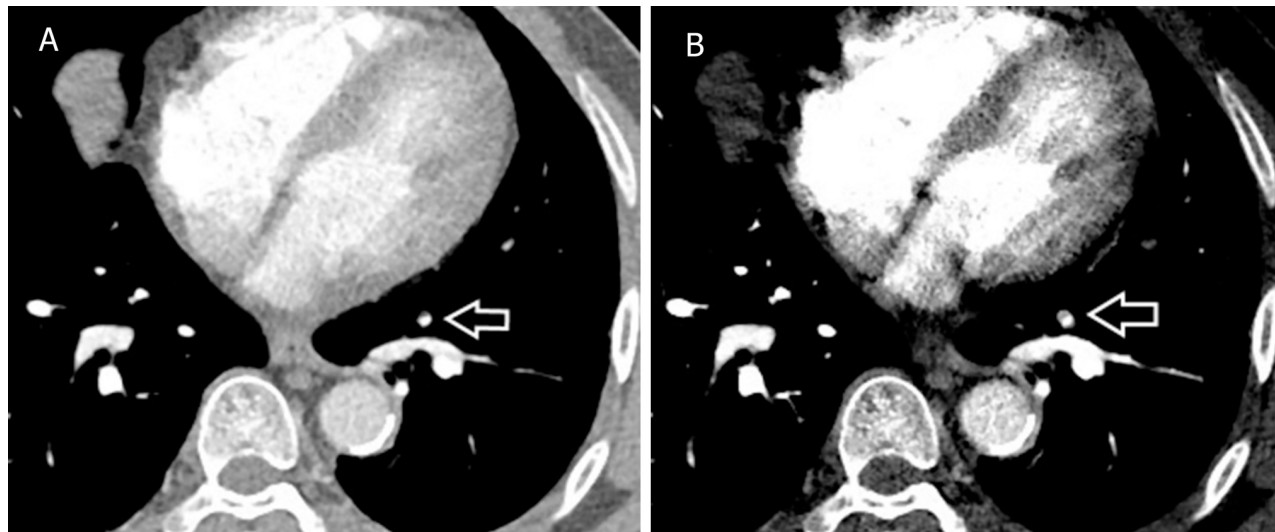

**Fig 5. Sample of an overlooked case.** A 22-year-old man with acute pulmonary embolism. DSDECT images at the level of the mid heart are displayed. A small left lower lobe segmental embolus (arrow) was overlooked in the standard mixed image (A) but was detected in the low-energy VMI+ (B). DSDECT, dual source dual energy computed tomography; VMI+, virtual monoenergetic plus.

**Table 4. Diagnostic index of the 40 keV VMI+ according to the location.**

| Location | Sensitivity | Specificity | AuROC | LR+ | PPV | NPV |
|---|---|---|---|---|---|---|
| MPA | 100 (2.5–100) | 100 (94.3–100) | 1 | - | 100 (2.5–100) | 100 (94.3–100) |
| RPA | 100 (76.8–100) | 98 (89.4–99.9) | 0.990 (0.970–1.) | 50.0 (7.9–348.0) | 93.3 (68.1–99.8) | 100 (92.7–100) |
| LPA | 90.9 (58.7–99.8) | 100 (93.3–100) | 0.955 (0.865–1) | - | 100 (69.2–100) | 98.1 (90.1–100) |
| RUL | 81.0 (58.1–94.6) | 95.3 (84.2–99.4) | 0.882 (0.790–0.973) | 17.4 (4.4–68.4) | 89.5 (66.9–98.7) | 91.1 (78.8–97.5) |
| RML | 90.5 (69.6–98.8) | 97.7 (87.7–99.9) | 0.941 (0.873–1) | 38.9 (5.6–271.0) | 95.0 (75.1–99.9) | 95.5 (84.5–99.4) |
| RLL | 90.5 (69.6–98.8) | 88.4 (74.9–96.1) | 0.894 (0.814–0.975) | 7.8 (3.4–17.9) | 79.2 (57.8–92.9) | 95.0 (83.1–99.4) |
| LUL | 100 (81.5–100) | 89.1 (76.4–96.4) | 0.946 (0.900–0.991) | 9.2 (4.0–21.0) | 78.3 (56.3–92.5) | 100 (91.4–100) |
| LLL | 100 (83.9–100) | 93.0 (80.9–98.5) | 0.965 (0.927–1) | 14.3 (4.8–42.7) | 87.5 (67.6–97.3) | 100 (91.2–100) |

Note (95% confidence interval)

The 40 keV VMI+, the 40 Kilo Electron Volt virtual monoenergetic plus; MPA, main pulmonary artery; RPA, right pulmonary artery; LPA, left pulmonary artery; RUL, right upper lobe; RML, right middle lobe; RLL, right lower lobe; LUL, left upper lobe; LLL, left lower lobe; AuROC, area under receiver operating characteristic; LR+, positive of likelihood ratio; PPV, positive predictive value; NPV, negative predictive value.

pulmonary arteries (Kappa > 0.90). Nevertheless, this correlation was slightly diminished in the assessment of the peripheral arteries (Kappa 0.76–0.90). This is possibly due to the small size of the peripheral vessels, the greater respiratory motion of the peripheral lung and the fact that the peripheral arteries were subject to partial volume effects between the vascular lumen and the adjacent lung parenchyma. Schueller-Weidekamm et al. also postulated that the difference in enhancement between the high and low energy protocols was larger for the peripheral pulmonary arteries in comparison to that of the central pulmonary arteries although there was no significant impact of the kilovoltage on the visualization of the segmental arteries [30].

Four locations (0.78%) were interpreted as negative for PE in the low-energy VMI+ dataset but positive in the standard mixed CT images. We found that one of these was located in the left pulmonary artery bifurcation, resulting in a small faint intraluminal hypodensity in the distal left pulmonary artery due to a flow artifact mimicking a thrombus seen in the standard mixed CT images. The remainder of these cases were sited in the distal segmental arteries an area with a greater degree of difficulty to analyze even with the standard mixed CT images.

On the contrary, twelve locations (1.95%) showed hypodense filling defects in the low-energy VMI+ dataset but had been interpreted as negative from the standard mixed CT images. Interestingly, all of the 12 locations were reviewed once again by the two radiologists which resulted in a consensus and it was noticed that each location contained a hypodense filling defect but had been overlooked or equivocally diagnosed from the standard mixed CT images. This encounter suggests that the low-energy VMI+ can improve the diagnostic value in doubtful cases or can enhance detection in the disregarded cases. These results are comparable to those of the earlier studies which claimed that improved visualization of the contrast-enhanced pulmonary arteries can be achieved with low-energy CT images [12, 13, 18, 27, 29, 30]. The greater iodine attenuation, the more significant demarcation of the hypodense embolism. The findings of this study lead us to suggest that low-energy VMI+ reconstructions should be included as part of the routine dual energy CTPA protocol to facilitate more accurate detection of acute PE.

Our study had some limitations. The assessment of acute PE in this study only included a comparison between the 40 keV VMI+ reconstructions and the standard mixed images. Other energy levels were not included in our study; nevertheless, previous studies had proposed the highest iodine contrast-to-noise ratios and the best image quality of the pulmonary artery circulation at 40 keV VMI+, hence our decision [21, 22, 27]. The body weight of the patients has

some influence on the low-energy imaging quality [31] but we did not concern ourselves with this in this study and could be included in future investigations. This study did not document the cardiac status of the patients which can have a considerable effect on the vascular enhancement. Finally, only subjective indicators of the image interpretation were determined without correlation with the clinical details or other laboratory findings.

## Conclusion

Low energy virtual monoenergetic plus DSDECT pulmonary angiography images are beneficial for improving the detection and diagnostic value of acute pulmonary embolism in the doubtful or disregarded routine standard mixed DSDECT images. Further studies with a larger sample size are warranted to support the results of this study.

## Supporting information

**S1 Data.**
(XLSX)

## Author Contributions

**Conceptualization:** Yutthaphan Wannasopha, Kantheera Leesmidt, Tanop Srisuwan, Juntima Euathrongchit, Apichat Tantraworasin.

**Data curation:** Yutthaphan Wannasopha, Kantheera Leesmidt, Tanop Srisuwan, Apichat Tantraworasin.

**Formal analysis:** Apichat Tantraworasin.

**Investigation:** Yutthaphan Wannasopha, Kantheera Leesmidt, Tanop Srisuwan, Juntima Euathrongchit.

**Methodology:** Yutthaphan Wannasopha, Kantheera Leesmidt, Tanop Srisuwan, Juntima Euathrongchit, Apichat Tantraworasin.

**Project administration:** Yutthaphan Wannasopha.

**Software:** Apichat Tantraworasin.

**Supervision:** Tanop Srisuwan, Juntima Euathrongchit, Apichat Tantraworasin.

**Validation:** Yutthaphan Wannasopha, Kantheera Leesmidt, Tanop Srisuwan, Juntima Euathrongchit, Apichat Tantraworasin.

**Visualization:** Yutthaphan Wannasopha, Kantheera Leesmidt, Tanop Srisuwan, Juntima Euathrongchit, Apichat Tantraworasin.

**Writing – original draft:** Yutthaphan Wannasopha, Kantheera Leesmidt, Tanop Srisuwan, Juntima Euathrongchit, Apichat Tantraworasin.

**Writing – review & editing:** Yutthaphan Wannasopha, Kantheera Leesmidt, Tanop Srisuwan, Juntima Euathrongchit, Apichat Tantraworasin.

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
