## [Decision Letter · Decision Letter 0]

6 Sep 2022

PONE-D-22-18744Value of Low-keV Virtual Monoenergetic Plus Dual-energy Computed Tomographic Imaging for Detection of Acute Pulmonary EmbolismPLOS ONE

Dear Dr. Tantraworasin,

Thank you for submitting your manuscript to PLOS ONE. After careful consideration, we feel that it has merit but does not fully meet PLOS ONE’s publication criteria as it currently stands. Therefore, we invite you to submit a revised version of the manuscript that addresses the points raised during the review process.

We look forward to receiving your revised manuscript.

Kind regards,

Alok Arora, MD, FACP, FRCP

Academic Editor

PLOS ONE

Journal Requirements:

3. Please include your tables as part of your main manuscript and remove the individual files. Please note that supplementary tables (should remain/ be uploaded) as separate "supporting information" files

Additional Editor Comments:

Please review comments and respond appropriately

Reviewers' comments:

Reviewer's Responses to Questions

**Comments to the Author**

1. Is the manuscript technically sound, and do the data support the conclusions?

Reviewer #1: Yes

Reviewer #2: Yes

2. Has the statistical analysis been performed appropriately and rigorously? 

Reviewer #1: Yes

Reviewer #2: Yes

3. Have the authors made all data underlying the findings in their manuscript fully available?

Reviewer #1: Yes

Reviewer #2: Yes

4. Is the manuscript presented in an intelligible fashion and written in standard English?

Reviewer #1: Yes

Reviewer #2: Yes

5. Review Comments to the Author

Reviewer #1: This is retrospective study comparing diagnostic value between standard protocol and DSDECT pulmonary angiogram for PE.

The manuscript is well written and analyzed. The conclusion is that DSDECT has better in detecting PE.

Since this is retrospective observational study, there could be observational bias. However, this study provides foundation for further studies with large sample size and more reviewers.

Reviewer #2: The manuscript does an excellent job of comparing between low energy VMI+ and standard dual energy CT for accurately diagnosing acute pulmonary embolism. It’s a well written and well represented manuscript. Even though the sample size is small (64 patient’s) authors did a great work in presenting their findings. Generalizability and validating the superiority of low energy VMI needs to be undertaken. Would advice authors to provide further information:

1) Any impact of BMI on diagnosing acute PE in the current sample

2) Advice authors to add BMI, h/o heart failure and possible sub group analysis for obese vs normal bmi group

3) For figure 1A and 1B please use arrow mark to point area of interest

6. PLOS authors have the option to publish the peer review history of their article (what does this mean?). If published, this will include your full peer review and any attached files.

Reviewer #1: **Yes: **Ashwini V Mallad

Reviewer #2: No

---

## [Author Response · Author response to Decision Letter 0]

15 Oct 2022

Dear Editors and Reviewers,

Thank you for allowing us to submit a revised version of our manuscript entitled “Value of Low-keV Virtual Monoenergetic Plus Dual-energy Computed Tomographic Imaging for Detection of Acute Pulmonary Embolism” for publication in “PLOS ONE”. We also thank the reviewers for their valuable comments that helped us improve our manuscript.

We have modified our manuscript incorporating the reviewers’ comments and we enclosed our response in a point-by-point manner. 

We hope the revised manuscript will now be found suitable for publication in “PLOS ONE”. We look forward to hearing from you soon.

Reviewer #1: This is retrospective study comparing diagnostic value between standard protocol and DSDECT pulmonary angiogram for PE.

The manuscript is well written and analyzed. The conclusion is that DSDECT has better in detecting PE.

Since this is retrospective observational study, there could be observational bias. However, this study provides foundation for further studies with large sample size and more reviewers.

Answer: Thank you very much. 

Reviewer #2: The manuscript does an excellent job of comparing between low energy VMI+ and standard dual energy CT for accurately diagnosing acute pulmonary embolism. It’s a well written and well represented manuscript. Even though the sample size is small (64 patient’s) authors did a great work in presenting their findings. Generalizability and validating the superiority of low energy VMI needs to be undertaken.

Would advice authors to provide further information:

1) Any impact of BMI on diagnosing acute PE in the current sample

Answer: No statistically impact of the BMI on the diagnosing acute PE in this current sample. This has been inserted in the result and the table 1. 

2) Advice authors to add BMI, h/o heart failure and possible subgroup analysis for obese vs normal

bmi group

 Answer: According to subgroup BMI analysis. There was no statistically significant difference between the positive and negative study groups in every BMI subgroup. This has been inserted in the result and the table 1. For the heart failure, we did not document the cardiac status of the patients in this study, and this was one of the limitation of our study. 

 3) For figure 1A and 1B please use arrow mark to point area of interest

 Answer: For the figure 1A and 1B we would like to show the overall imaging quality comparison between the standard mixed image and the low-energy VMI+ image. The arrows have been inserted in the figures already.

---

## [Editor Report · Decision Letter 1]

19 Oct 2022

Value of Low-keV Virtual Monoenergetic Plus Dual-energy Computed Tomographic Imaging for Detection of Acute Pulmonary Embolism

PONE-D-22-18744R1

Dear Dr. Tantraworasin,

We’re pleased to inform you that your manuscript has been judged scientifically suitable for publication and will be formally accepted for publication once it meets all outstanding technical requirements.

Kind regards,

Alok Arora, MD, FACP, FRCP

Academic Editor

PLOS ONE
---

## [Editor Report · Acceptance letter]

2 Nov 2022

PONE-D-22-18744R1 

Value of Low-keV Virtual Monoenergetic Plus Dual-energy Computed Tomographic Imaging for Detection of Acute Pulmonary Embolism 

Dear Dr. Tantraworasin:

I'm pleased to inform you that your manuscript has been deemed suitable for publication in PLOS ONE. Congratulations! Your manuscript is now with our production department. 

Kind regards, 

on behalf of

Dr. Alok Arora 

Academic Editor

PLOS ONE